# A Genome-Scale Metabolic Model of *Thalassiosira pseudonana* CCMP 1335 for a Systems-Level Understanding of Its Metabolism and Biotechnological Potential

**DOI:** 10.3390/microorganisms8091396

**Published:** 2020-09-11

**Authors:** Ahmad Ahmad, Archana Tiwari, Shireesh Srivastava

**Affiliations:** 1Systems Biology for Biofuel Group, International Centre for Genetic Engineering and Biotechnology (ICGEB), New Delhi 110067, India; ahmadbioinfo@gmail.com; 2Department of Biotechnology, Noida International University (NIU), Noida 203201, India

**Keywords:** diatom, photosynthesis, flux analysis, fucoxanthin, heterologous products

## Abstract

*Thalassiosira pseudonana* is a transformable and biotechnologically promising model diatom with an ability to synthesise nutraceuticals such as fucoxanthin and store a significant amount of polyglucans and lipids including omega-3 fatty acids. While it was the first diatom to be sequenced, a systems-level analysis of its metabolism has not been done yet. This work presents first comprehensive, compartmentalized, and functional genome-scale metabolic model of the marine diatom *Thalassiosira pseudonana* CCMP 1335, which we have termed *i*Thaps987. The model includes 987 genes, 2477 reactions, and 2456 metabolites. Comparison with the model of another diatom *Phaeodactylum tricornutum* revealed presence of 183 unique enzymes (belonging primarily to amino acid, carbohydrate, and lipid metabolism) in *i*Thaps987. Model simulations showed a typical C3-type photosynthetic carbon fixation and suggested a preference of violaxanthin–diadinoxanthin pathway over violaxanthin–neoxanthin pathway for the production of fucoxanthin. Linear electron flow was found be active and cyclic electron flow was inactive under normal phototrophic conditions (unlike green algae and plants), validating the model predictions with previous reports. Investigation of the model for the potential of *Thalassiosira pseudonana* CCMP 1335 to produce other industrially useful compounds suggest iso-butanol as a foreign compound that can be synthesized by a single-gene addition. This work provides novel insights about the metabolism and potential of the organism and will be helpful to further investigate its metabolism and devise metabolic engineering strategies for the production of various compounds.

## 1. Introduction

Diatoms are unique unicellular photosynthetic eukaryotic microbes that have silica incorporated in their cell wall [1]. They play a significant role in global carbon and silicon recycling [1] and contribute about half of aquatic primary production and a quarter of the total primary productivity [2]. They are attractive organisms that can be exploited as cell factories to convert atmospheric CO_2_ into a wide range of industrially and medically useful compounds [3,4,5]. Diatoms primarily store glucose in the form chrysolaminarin [6], a polyglucan (primarily composed of β(1→3) linked glucose monomers) which can be used as renewable feedstock for fermentation and subsequent production of platform chemicals. Furthermore, their ability to store a significant amount of lipids (20–30% and in some species up to 45–60% of the dry cell weight) [7] and synthesize nutraceuticals such as omega-3 fatty acids [7,8] and fucoxanthins [3,9] makes them attractive organisms. 

*Thalassiosira pseudonana* is a model diatom. It was the first diatom chosen for genome sequencing due to its utility in physiological and silica biomineralization [10] studies. It is also considered to be a potential drug delivery vehicle for cancer treatment during chemotherapy [11]. The ability to grow in marine water makes it an attractive organism for large-scale cultures needed for the bioenergy applications, as it will not compete with freshwater and land. Unsurprisingly, various studies have applied genetic engineering in *T. pseudonana* for enhanced production/accumulation of lipids, TAGs, and omega-3 fatty acids [12,13,14].

Genome-scale metabolic models (GEMs) make it possible to simulate metabolic behavior of the organism and predict internal metabolism under different conditions [15,16,17]. The GEMs contain a majority of the metabolic reactions present in an organism based on its annotated genome sequence [18,19]. A large number of GEMs of a variety of organisms ranging from prokaryotes [20,21] to eukaryotes [15,22,23] have been reconstructed and utilized to analyze their metabolic behavior. Moreover, the GEMs of photosynthetic organisms, e.g., microalgae (*Chlamydomonas* [24,25], *Chlorella* [26], and *Nannochloropsis* [15,27]) and the diatom (*Phaeodactylum tricornutum* [28,29]) have been reconstructed and used to analyze the process of photosynthesis and metabolism under light and dark conditions. While there are two published GEMs available for *P. tricornutum*, a GEM for *T. pseudonana* is still not available. *P. tricornutum* and *T. pseudonana* have been segregated under taxonomically distinct subgroups (order and genus) [30,31]. They have been kept in two different groups, pennate, and centric for *P. tricornutum* and *T. pseudonana*, respectively [32], based on arrangement of frustules on their cell wall. Si^+^ was found to be an essential nutritional component for the growth of *T. pseudonana* [33] while it was non-essential for the growth of *P. tricornutum* [34,35]. Moreover, under phosphorus (P) stress, genes related to Calvin cycle, fatty acid biosynthesis, and *PEPCK* (phosphoenolpyruvate carboxykinase) show differential regulation in *T. pseudonana* and *P. tricornutum* [33]. Thus, a GEM of *P. tricornutum* cannot be used to investigate the metabolism and biotechnological potential of a significantly different organism, *T. pseudonana,* as suggested by a recent work where eight different strains of the same organism (*E. coli*) were found to be physiologically different [21]. Therefore, it is imperative to work on the GEM of the same strain to perform systems level studies.

In this work, we present the first high-quality manually-curated GEM (*i*Thaps987) of *T. pseudonana*, to overcome this gap and perform its systems level investigation. The model *i*Thaps987 includes 987 genes, 2477 reactions, and 2456 metabolites. The FBA analysis revealed the CO_2_ fixation by C3/Calvin–Benson–Bassham (CBB) cycle. The different pathway routes for biosynthesis of fucoxanthin were explored for their activity and the results indicate a preference for the violaxanthin–diadinoxanthin pathway over the violaxanthin–neoxanthin pathway. The features of this model were compared to the previously published *P. tricornutum* model (*i*LB1025) [28]. The work also focuses on the potential of *T. pseudonana* to photosynthetically produce some industrially useful compounds. 

## 2. Materials and Methods 

### 2.1. Reconstruction of a Genome-Scale Metabolic Model

The genome-scale metabolic model of *Thalassiosira pseudonana, i*Thaps987, was reconstructed and analysed using established protocols and an approach described for a number other organisms such as *Geobacillus thermoglucosidasius* [20], *Salmonella typhimurium* [17], and *Arabidopsis thaliana* [36]. The model was reconstructed using the python-based metabolic modeling tool ‘ScrumPy’ [37]. The model was reconstructed based on organism-specific data obtained from BioCyc [38] database and the annotated genome sequence from the National Center for Biotechnology Information (NCBI) database. The macromolecular composition of *Thalassiosira pseudonana* biomass was measured to construct the biomass reaction which provides molar fractions of metabolites required to make a gram of biomass. Information from various biological database were used as per requirement. For example, reaction directionalities were taken from BioCyc [38] and MetaCyc [39] database and the metabolite formulae that were missing in Biocyc database were taken from KEGG [40] database or from previously published models [15,20,41].

### 2.2. Draft Model

The Pathway Genome Database (PGDB) for *Thalassiosira pseudonana* (Thaps, v19.5) was downloaded from BioCyc and extracted to obtain a flat-files database. The PGDB of *Thalassiosira pseudonana* has information related to reactions, their genes and sequences. ScrumPy tool was used to fetch these flat-file databases using the inbuilt module ‘PyoCyc’. Finally, a draft model was created by obtaining all the reactions of the database.

The initial draft model included several non-specific metabolites, e.g., ‘Aldoses’, ‘Fatty aldehydes’, and reactions—e.g., ‘DNA-LIGASE’—that would create uncertainties in the model. The draft model was refined iteratively using literature, biological database(s), and previously published models until a final, robust model is obtained. In the first round of refinement, the non-specific metabolites and reactions were removed from the model. In the subsequent step, each reaction was checked and corrected, wherever required, for mass balance in terms of carbon, nitrogen, phosphorus, and sulfur. Further refinements were carried out as explained below.

### 2.3. Gap-Filling of the Model

The draft model obtained after the initial refining was unable to synthesise some essential biomass precursors, indicating an absence of at least one key reaction that participates in the synthesis of the corresponding precursor. The absence of such key reactions results in gaps in the network. These gaps in the draft model were manually identified by performing optimisations for production of all the biomass precursors one by one. Those precursors which were unable to be produced by the draft model were termed as gapped metabolites. The presence of gapped metabolites in a model indicates existence of some missing links (reactions/pathways), also known as gapped reactions/pathways, in the network. These gaps were filled manually by searching for biosynthetic pathways and reactions that will ensure biosynthesis of these biomass precursors, thereby filling those gaps in the network. The gene sequences of these added reactions were subjected to BLAST searches against the *Thalassiosira pseudonana* genome sequence.

### 2.4. Compartments in the Model

*Thalassiosira pseudonana* is a eukaryotic microbe and has different compartments for different functions. All reactions of the model were localized into three different compartments: cytosol, chloroplast, and mitochondrion. The sub-cellular localisation of reactions was obtained using ngLoc web-server [42]. A reaction was assigned to cytosol if no subcellular localization could be obtained.

### 2.5. Energy Requirements

A characteristic feature of all organisms is to grow and maintain their structural integrity. They require energy to perform these two basic functions of life. This energy is required in the form of the cellular currency ‘ATP’. The energy (ATP) required for growth is termed as growth-associated ATP maintenance (GAM). The GAM accounts for energy required to synthesize precursors, polymerize purine and pyrimidine bases and amino acids to form DNA and proteins, respectively, and polymerize cell wall precursors, etc. The other type of energy (ATP) which accounts to maintain cellular structure and integrity is termed Non growth-associated ATP maintenance (NGAM). As the values of GAM and NGAM are not available for this strain, the values from the GEM of *Phaeodactylum tricornutum* [29] were used. Similar assumptions are regularly employed in cases where the actual values are not available. For example, the GEM of *Synechococcus* sp. BDU130192 had utilized ATP maintenance values from previously published models of *Synechococcus* sp. PCC 7002 [41], while the GEMs of *Thermus thermophilus* HB27 [43], and tomato (*Solanum lycopersicum* L.) [44] have utilized ATP maintenance values from previously published models of *E.coli* and *Arabidopsis thaliana* models, respectively. We have also evaluated the effect of variation of NGAM on intracellular flux distributions.

### 2.6. Measurement of Biomass Composition for Setting Up the Biomass Equation

The biomass equation was formulated by experimental measurement of important biomass components like proteins, carbohydrates, lipids, DNA, and RNA while the minor components—e.g., pigments—were taken from previously published models [28]. Further details on the methods used to measure the biomass composition are provided in Appendix A.

### 2.7. Model Simulations 

The genome scale metabolic model, *i*Thaps987, representing the metabolic reactions of *T. pseudonana* was reconstructed as a ScrumPy-readable text file (Appendix A) as well as mat format (Appendix A) and is compatible with COBRA Toolbox and ScrumPy modeling packages. An Excel file of the model is also provided (Appendix A). 

The model was analyzed using ‘Flux Balance Analysis’ (FBA), a constraint-based modeling (CBM) approach based on linear programming technique using ScrumPy. FBA determines the internal flux distribution which satisfies the steady state assumption besides some other constraints. According to the steady state assumption, the concentrations of intracellular metabolites remain steady (i.e., they do not vary with time). This implies that the compounds taken in by the cell should either go towards biomass production or towards the formation some byproducts. Overall, the carbon, nitrogen, etc. should be balanced in the determined flux distributions. The FBA approach optimizes any metabolic property, e.g., biomass formation reaction, or other reaction(s) within the constraints provided. In this study, minimization of total cellular fluxes was used as the cellular objective function [20]. For simulating photoautotrophic conditions, biomass reaction (growth rate) was fixed at the experimentally measured value of 0.024 h^−1^ (see Appendix A) while the import and export of nutrients including CO_2_ and photon intake were left free. The linear programming problem for minimization of total fluxes was formulated as
$$\mathrm{Minimize}\;\sum_{i=1}^{n}v_{i}$$
subject to S.v=0
$$v_{\mathrm{Biomass}}\;=\;t$$
$$v_{\mathrm{ATP\_Maintenance}}\;=\;\mathrm{ATP}\;\mathrm{Maintenance}$$
where *S* is the stoichiometry matrix, v is the reactions flux vector, n is number of reactions,$\;v_{\mathrm{Biomass}}$ is the flux through the biomass reaction, t is some positive value (measured growth rate) and *i* is iteration.

### 2.8. Simulations of Si^+^ Limited Conditions

In order to analyse the effect of silica limitation on chrysolaminarin production, a virtual transport reaction was added to the model for transport of chrysolaminarin from cytosol to an ‘extracellular’ compartment. That is, the accumulation of chrysolaminarin is assumed as ‘export’. The CO_2_ intake rate was fixed to 2.05 mmol/(gDCW.h) while the silica uptake rate was varied from 0 mmol/(gDCW.h) to 0.05 mmol/(gDCW.h) and minimization of total flux was used as the objective function.

### 2.9. Reaction Essentiality Analysis

The FBA approach (with minimization of total fluxes as objective) was used to perform the reaction essentiality analysis. The flux through each reaction was fixed to zero at a time while the flux through biomass reaction (growth rate) was fixed to the experimentally observed value at the time of simulations. A reaction was defined as essential if fixing the flux through the reaction to zero results in an infeasible optimization.
$$\mathrm{Minimize}\;\sum_{i=1}^{n}v_{i}$$
subject to S.v=0
$$v_{\mathrm{Biomass}}\;=\;t$$
$$v_{j}=0$$
$$v_{\mathrm{ATP\_Maintenance}}\;=\;\mathrm{ATP}\;\mathrm{Maintenance}$$
where, *S* is the stoichiometry matrix, v is the reactions flux vector, $v_{j}$ is the reaction that is being ‘deleted’, n is number of reactions,$\;v_{\mathrm{Biomass}}$ is the flux through the biomass reaction, t is some positive value (measured growth rate) and $v_{i}$ is i^th^ value in flux vector.

### 2.10. Maximum Theoretical Yields for Various Industrial Compounds

Diatoms can be exploited as cell factories [4]. Therefore, we explored the model for its potential to produce various native and non-native compounds. The maximum theoretical yield for each product was predicted by: (i) constraining the biomass reactions to 80% of its wild-type growth rate, (ii) setting the exchange reaction of the corresponding product as the objective function. The transport and exchange reactions were added wherever required to the model. 

The maximum theoretical yields for the heterologous products were predicted by adding the required external reactions and constraining the model as done for indigenous products. The MetaCyc database contains pathways and their respective reactions as mini modules, e.g., pathway for ‘styrene’ biosynthesis contains specific reactions that synthesises ‘styrene’. Using an in-house script, the presence of these specific reactions was checked in *i*Thaps987, and the missing reactions were ‘temporarily’ added to the model. 

The linear programming formulation to perform FBA for the production of the native and heterologous products is
$$\mathrm{Maximise}\;\sum_{i}^{n}c_{i}.v_{i}$$
subject to S.v=0
$$v_{\mathrm{Biomass}}\;=\;t$$
$$v_{\mathrm{ATP\_Maintenance}}\;=\;\mathrm{ATP}\;\mathrm{Maintenance}$$
$$\mathrm{Max}\;\mathrm{Yield}\;(x)=\frac{v_{obj}}{v_{c}}$$
where *S* is the stoichiometry matrix, v is the reactions flux vector and *c* is the objective function. In this case, the objective function is the exchange reaction(s) for product formation. $v_{obj}$ and $v_{c}$ are the fluxes through the objective function and the carbon source, respectively and x is maximum theoretical yield of the target product. Note that the biomass production flux *v*_Biomass_ was kept as 80% of the measured growth rate. The maximum yields (x) were obtained as a ratio of product formation to carbon intake rates. 

## 3. Results

### 3.1. Gap Filling

The draft model was subjected to manual gap finding analysis, which identified only 62 orphan enzymes additionally required (absent in the *T. pseudonana* genome) for biomass-precursor biosynthesis. Out of these 62 enzymes, 21 were already present in the PGDB of *T. pseudonana* while the 14 enzymes were found in *P. tricornutum* or *Chlamydomonas reinhardtii* genomes. Also, there are 19 orphan metabolic reactions present in the model that do not have any associated Enzyme Commission (EC) number. A list of gap-filling reactions is summarised in Appendix A.

### 3.2. General Properties of the Model

The final model, *i*Thaps987, contains 987 genes, 2477 reactions, and 2456 metabolites. The model comprises of 2389 metabolic, and 88 transport and exchange reactions. The model has three compartments, namely cytosol, chloroplast, and mitochondrion. A total 1125 reactions operate in cytosol while 887 are present in chloroplast and 377 reactions are located in mitochondrion (Table 1). In terms of metabolites, 1024 metabolites are present in cytosol followed by 984 in chloroplast and 467 in mitochondrion (Table 1).

The model *i*Thaps987 is the second largest in terms of the number of reactions compared to other publicly available models of diatoms (Table 2). 

### 3.3. Comparison of i*Thaps987* Model with P. tricornutum Model, i*LB1025*

The *i*Thaps987 model was compared with *i*LB1025, the GEM of another diatom, *P. tricornutum*, to investigate similarity and differences between the two models. The models were compared primarily based on Enzyme Commission (EC) numbers. The comparison revealed 483 enzymes are common in both models (see Appendix A) and 183 enzymes are unique in *i*Thaps987 and 191 in *i*LB1025, respectively (Figure 1a). Further analysis of the common enzymes revealed that enzymes related to amino acid metabolism were the most common between the two models (128), followed by the enzymes related to fatty acid (79) and carbohydrate (70) metabolism, respectively. The enzymes related to vitamins and cofactor (59), nucleotide and carotenoid (34) metabolism also had a significant number of common enzymes. There were other pathways which had slightly fewer enzymes common to both models—e.g., nitrogen metabolism, oxidative phosphorylation, and photosynthesis. Appendix A contains comparison of enzymes of both models. Furthermore, the analysis of these 183 unique enzymes present in *i*Thaps987 revealed that most of the unique enzymes participate in carbohydrate and amino acids metabolism (40 and 38 respectively), followed by lipid metabolism pathways with 29 enzymes. Pathways related to vitamins and co-factor metabolism, pigments, and nucleotides have relatively fewer unique enzymes (Figure 1b). 

Figure 1c shows the distribution of unique enzymes in sub-pathways of main pathway categories. Among the 38 unique enzymes that participate in amino acid metabolism, most relate to arginine and proline metabolism, followed by glycine, serine, and threonine metabolism and phenylalanine, tyrosine, and tryptophan biosynthesis. Regarding the carbohydrate metabolism, pyruvate metabolism has highest number of unique enzymes (9) followed by glycolysis (8), fructose-mannose metabolism (5), and pentose phosphate pathways (4). The nine unique enzymes of pyruvate metabolism are phosphoenolpyruvate synthase, lactoylglutathione lyase, pyruvate dehydrogenase (quinone), propionate CoA-transferase, hydroxyacylglutathione hydrolase, pyruvate formate-lyase, malate dehydrogenase (oxaloacetate-decarboxylating), L-lactate dehydrogenase (cytochrome), and phosphoenolpyruvate carboxykinase (diphosphate). Sub-pathways of lipids metabolism- glycerophospholipid (8) and fatty acid (8) metabolism have equal number of unique enzymes while arachidonic acid (3) and glycerolipid (1) pathways have lesser number of enzymes. Among the vitamins and co-factor metabolism pathways, almost all sub-pathways have approximately the same number of unique enzymes. Porphyrin and chlorophyll (6) and carotenoids (3) metabolism pathways have more unique enzymes than other sub-pathways of the pigment metabolism pathway. The Appendix A has a list of all unique enzymes present in *T. pseudonana*.

### 3.4. Model Simulations 

For the photoautotrophic simulations, the growth rate was fixed to the experimentally determined value of 0.024 h^−1^ (see Appendix A for the growth curve). The CO_2_ and photon uptake rates were predicted to be 1.05 and 13.02 mmol/(gDCW.h), respectively. These uptake rates are within the physiological ranges suggested by previous reports [15,28]. The nitrate uptake rate was found to be 0.19 mmol/(gDCW.h). The model simulations predicted O_2_ evolution of 1.51 mmol/(gDCW.h). Thus, the O_2_/CO_2_ ratio was predicted to be 1.43 which is very close to the value 1.37 obtained with the *Phaeodactylum tricornutum* model *i*LB1025 [28] for the same growth rate. 

### 3.5. Light-Driven Generation of ATP and NADPH: Light Reaction

The light-driven growth simulations (photoautotrophic growth) showed CO_2_ fixation by the C3 cycle by utilising ATP and NADPH produced during light reactions in chloroplast. The photon directs the photolysis of water and electron flow from water to NADPH involving both photosystems PSII and PSI.

During the electron flow, protons are pumped in thylakoid lumen which generates an electrochemical gradient across the thylakoid membrane. This gradient provides the force (proton motive force) for plastidial ATP synthesis by ATP synthase complex. The flux distribution reveals that under normal photoautotrophic condition, the cyclic electron flow (CEF), which only allows production of ATP, is inactive. Flux through the linear electron flow (LEF), that facilitates simultaneous and coupled biosynthesis of ATP and NADPH, was found to be 3.4 mmol/(gDCW.h).

### 3.6. Carbon Fixation by the CBB Cycle (C3 Cycle): Dark Reaction

CO_2_ is fixed by the enzyme Rubisco that converts three molecules of CO_2_ and three molecules of ribulose 1,5-bisphosphate (RUBP) into six molecules of 3-phosphoglycerate (3PGA) which is finally reduced to glyceraldehyde-3-phosphate (GAP). The carbon flux is mainly distributed through GAP to three different routes—i.e., the CBB cycle for the regeneration of RUBP—the pentose phosphate pathway and the upper part of glycolysis in chloroplast. About 0.3% of the carbon goes to lower part of glycolysis to form PEP, PYR, and acetyl CoA while 6.7% forms hydroxypyruvate. The rest 92.7% of the carbon flux goes to form GAP (Figure 2). The carbon from GAP is further diverged into two routes—firstly, towards fructose 1,6-DP formation that leads to synthesis of storage carbohydrate and secondly towards the pentose phosphate pathway (PPP) which leads to recycling of RUBP formation in C3 cycle.

In chloroplast, the acetyl-CoA initiates fatty acid synthesis [28,33,45] using the ATP and NAD(P)H generated during photosynthesis. The fluxes through fatty acid synthesis reactions were found to be very low. The carotenoids biosynthesis also takes place in chloroplast [46]. Phosphoenol pyruvate (PEP) and pyruvate form farnesyl pyrophosphate (FPP) through a series of reactions and finally the FPP gets converted through a cascade of reactions into carotenoids like beta-carotenoids, astaxanthin, etc. A very small amount of flux goes through these carotenoid biosynthesis reactions.

Interestingly, the model predicts that the pyruvate that takes part in TCA cycle in mitochondrion comes from chloroplast via cytosol through inter-compartmental transporters. The pyruvate gets converted into acetyl-CoA which participates in the TCA (tricarboxylic acid cycle). The flux through pyruvate to alpha-ketoglutarate is 0.03 mmol/(gDCW.h) (Figure 2). Although a complete TCA cycle is present in model, only a few initial and final reactions carry fluxes. The partial activity of the TCA cycle in diatoms has also been reported in previous studies [29].

Urea cycle was found to be partially active under normal growth conditions. The flux through carbamoyl phosphate synthase, which converts ammonium and carbonate ions (inside mitochondrion) to carbamoyl phosphate, was found to be 0.004 mmol/(gDCW.h). Carbamoyl-phosphate then reacts with l-ornithine to form l-citrulline which is then transported to cytoplasm. In cytoplasm, l-citrulline reacts with l-aspartate to form arginosuccinate which, in turn, dissociates into arginine and fumarate. The flux through all above reactions is same (0.004 mmol/(gDCW.h)). The last reaction of the urea cycle, which converts arginine to urea and l-ornithine, is not active under normal conditions. Only upon increasing the nitrate uptake rate without increasing the growth rate is this reaction activated.

### 3.7. Simulation of Si^+^ Limited Condition

According to previous reports, *T. pseudonana* accumulates a significant (20–25%) amount of chrysolaminarin [47]. In order to analyze the effect of silica limitation on chrysolaminarin production, Si^+^ transport was varied and the effect on the flux through chrysolaminarin production was calculated (see Section 2.8 for more details) [48]. The simulation results show that at very low flux of silica transport, chrysolaminarin production is the highest and biomass production is the lowest (Figure 3). It can be observed that with an increase in silica transport, biomass formation flux increases, and chrysolaminarin production decreases.

A comparison of the flux distribution under Si-replete and Si-limiting conditions revealed reduced photon intake and O_2_ released for the same CO_2_ intake rate (Figure 2). The reduced O_2_ production under Si^+^ limited conditions is in agreement with a previous study in *Thalassiosira weissflogii* [49]. That study has shown a reduced activity of PSII, the site of O_2_ production, under Si^+^-limiting condition. Similarly, the reduced photon intake under Si^+^ limitation is also in agreement with a previous report [49]. Flux through all pathways was reduced upon Si-limitation, while carbohydrate-synthesis flux was significantly increased.

### 3.8. Effect of Maintenance Energy on Flux Predictions

We evaluated the effect of varying ATP maintenance values on metabolism. The range of ATPM chosen for the scanning analysis was 0–8 mmol/(gDCW.h). This range was chosen because the lowest ATPM value of 0.36 mmol/(gDCW.h) has been reported for *Lactobacillus planetarium* [28] while the highest, 7.6 mmol/(gDCW.h) [28], has been reported for *E. coli*. At ATPM value of zero, the ATP is synthesized primarily in chloroplast while a very small fraction of ATP is synthesized in mitochondrion. As the ATPM demand increases, both mitochondria as well as chloroplast contribute to ATP production. However, the increase in the mitochondrial ATP synthesis was simulated to be more in comparison to the increase in plastidial ATP synthesis (Figure 4a) and finally, the ratio of plastidial ATP synthesis to mitochondrial ATP synthesis reaches to about 0.8 (Figure 4a).

The effect of ATPM values on various reactions reveal that as ATPM increases, more photons are absorbed and the flux through linear electron flow via the PSII and PSI increases. The cyclic electron flow (CEF) remains inactive. The flux through oxaloacetate–malate valve was also increased (Figure 4b). Simulations performed while knocking out (fixing the flux to zero) the oxaloacetate–malate valve resulted in shifting the ATP production from mitochondria to chloroplast by activating the CEF (Figure 4c).

The results also show that as ATPM increases, more photons are absorbed, more O_2_ is consumed in mitochondria and more O_2_ is produced in the chloroplast. The O_2_ production increases with O_2_ consumption linearly (Appendix A). This trend of O_2_ production vs. O_2_ consumption is in agreement with previous reports [28,50].

### 3.9. Photoautotrophic Production of Industrially-Relevant Compounds

Diatoms are potential organisms for photoautotrophic production of various medically and industrially useful products [3,4,5,10]. They contain various valuable natural compounds like lipids, fucoxanthin pigments, etc. We investigated the potential of this organism to produce some native and non-native metabolites. The compound malate is used as a precursor for the production of biodegradable polymers, and in pharmaceutical, food, and beverages industries. Succinate is used as growth regulator in agriculture as well as precursor for many useful industrial compounds like 1,4-butanediol, tetrahydrofuran, and polybutylene succinate. Citrate is mainly used in food and pharmaceutical industries. Among the non-native compounds, 2-methyl succinate, styrene, and isobutanol are used as pharmaceutical, biopolyester, and biofuel, respectively.

The model *i*Thaps987 was used to investigate the potential of *T. pseudonana* metabolic network to produce various native and non-native compounds (Table 3). The theoretical yields (mol product produced/mol CO_2_ consumed) of these products were predicted by the model using the FBA approach by fixing the growth rate to 80% of the wild type growth rate and maximizing for product formation. Among the native metabolites, malate, and succinate have same yield of 0.22 while citrate has a yield of 0.14. Among non-native compounds, isobutanol and PHB (polyhydroxybutyrate), have a maximum yield of 0.22 followed by 2-methylsuccinate with 0.17 and styrene with 0.17. Additionally, isobutanol production should require the least metabolic engineering, i.e., introduction of only a single heterologous gene.

### 3.10. Reaction Deletion Analysis to Identify Essential Reactions

The reaction-deletion analysis revealed a total of 327 essential reactions under photoautotrophic condition. Figure 5 shows the distributions of total and essential reactions across different pathways. The fatty acid metabolism pathway has the highest numbers of essential reactions followed by the nucleotide and pigment metabolism pathways. The low percent of essential reactions in *T. pseudonana* across the metabolic pathways suggests the presence of alternative pathways/routes for the biosynthesis of biomass precursors.

## 4. Discussion

Diatoms are a major constituent of phytoplanktons and play a central role in marine primary productivity. They can also be exploited as potential feedstock for biotechnological processes. Although a considerable progress has been made in understanding their basic and molecular biology, much remains to be done in the domain of systems biology. In this report, a first genome-scale metabolic model of *T. pseudonana* was used to interrogate different aspects of metabolism and biotechnological potential of the *T. pseudonana* in a systems approach. The model *i*Thaps987 is mass balanced and does not contain any thermodynamically infeasible cycles.

During the course of model reconstruction, we identified a number of previously unannotated genes in *T. pseudonana*. Therefore, the model can be used for improvement of genome annotation. The draft model lacked the “PHOSACETYLTRANS-RXN” reaction but BLAST search revealed that locus “XP_002287976.1” codes for the same enzyme. The reactions “ISOCITRATE-DEHYDROGENASE-NAD+-RXN” is coded by the gene locus “XP_002290637.1”, “MALATE-DEHYDROGENASE-NADP+-RXN” by the gene locus “XP_002296311.1” and “MALATE--COA-LIGASE-RXN” by the gene locus “XP_002286405.1”. All of these gene loci were labelled as “hypothetical protein” or “predicted protein”.

Comparison with the metabolic model (*i*LB1025) of another diatom *P. tricornutum* revealed common and unique features of the model, *i*Thaps987 (see Section 3.3 and Appendix A for more details). It was interesting to note that a large number of unique enzymes in the *T. pseudonana* metabolic model are involved in pyruvate metabolism.

The simulations results predicted a typical C3 type carbon fixation (Figure 3) in spite of the presence of C4-type carbon-fixing enzyme genes. A previous study, using ^14^CO_2_ labelling, has shown that indeed C3 type carbon fixation is active under normal photoautotrophic condition [51]. The model prediction was very well in agreement with the previous study.

The ATP in the chloroplast can be generated either by the linear electron flow (LEF) or by the cyclic electron flow (CEF). The ATP synthesized by LEF is tightly coupled with NADPH production whereas the ATP synthesized by the CEF does not generate any NADPH. The model simulation indicates that the plastidial ATP was synthesized exclusively by LEF. This prediction is also in agreement with the experimental observation by [50] that the CEF around PSI is negligible in diatoms. Also, we performed the simulations by changing the photon uptake ratio between PSII/PSI to 2:1 and 4:1 to explore the effect of PSII/PSI ratio on flux distribution and specially on the flux through CEF. There was no significant change in the flux distributions under both conditions except that the flux through PSII was increased to 3.5 (mmol/(gDCW.h)) from 1.7 (mmol/(gDCW.h)) (almost doubled). CEF was still inactive.

The model was further utilized to explore the effect of different ATPM values on metabolism (Figure 4b). The increase in ATP demand at higher ATPM values causes plastidial ATP synthase to produce more ATP by absorbing more photons. The simulation reveals that even at higher ATPM value, only LEF is responsible for plastidial ATP synthesis (as shown by [50]) along-with NADPH production. This deviates the ATP/NADPH ratio. Mitochondrion plays an important role to optimize the disturbed ATP/NADPH ratio in chloroplast. The chloroplastic enzyme malate dehyrogenase converts oxaloacetate to malate by converting NADPH to NADP. The malate was then transported to mitochondrion via the malate–oxaloacetate valve (while oxaloacetate is shuttled to chloroplast). The mitochondrial enzyme malate dehydrogenase converts the malate back to oxaloacetate by converting NAD to NADH. This NADH is then utilized by mitochondrial ATP synthase to produce more ATP. The ratio of mitochondrial ATP to plastidial ATP production was close to zero at low ATPM (Figure 4a) but as ATPM increases, the ratio also increases. Therefore, the ATP/NADPH ratio was optimized by mitochondrial ATP synthesis. This is in agreement with [50] which found that a part of the photosynthate is channelled to mitochondrion in order to regulate the ATP/NADPH ratio in chloroplast. Only when we block the malate valve in mitochondrion, the CEF is activated. Further analysis revealed that blocking malate valve leads to a higher photon uptake and if we fix photon uptake to the previous value then solution becomes infeasible. This result indicates the important role played by the mitochondrion and the malate valve in optimizing photosynthesis (Figure 4c). The GEM, *i*Thaps987 has ‘2-oxoglutarate-malate shuttle’ besides ‘malate–oxaloacetate shuttle’ for exchange of metabolites between mitochondrion and chloroplast. Amongst these, the ‘malate–oxaloacetate shuttle’ plays a major role in translocation of photosynthate from chloroplast to mitochondrion. ‘Malate–aspartate shuttle’ was found to be absent in the *i*Thaps987 model as it was in *i*LB1025.

Unlike other unicellular organisms such as bacteria, cyanobacteria, yeast, and green algae, diatoms have a complete urea cycle [52]. However, model simulations suggest that under normal photoautotrophic conditions, urea cycle is only partially active. Only upon excessive nitrogen (NO_3_^−^) intake does the last reaction of urea cycle get activated to flush out the excess nitrogen in the form of urea.

The simulations for Si-limited condition revealed a trade-off between formation of storage product and biomass production under nutrient-limiting condition. The carbon was partitioning between biomass and chrysolaminarin at various levels of silica. Under stressed conditions (very low silica), most of the carbon goes to chrysolaminarin while at higher silica intake flux, the carbon flux shifts towards biomass formation.

The systems analyses of *i*Thaps987 with respect to its biotechnological potential revealed iso-butanol as the non-native product that can be produced with the introduction of only a single foreign gene “kivD’. On the contrary, two genes ‘kivD and ADH2′ are required in the *P. tricornutum* model.

The reconstructed GEM can also be utilized to reveal new systems-level insights about the organism. For example, integration of expression data to the available GEM could provide novel insights into metabolic regulation under different conditions. The model is provided in ScrumPy as well as in mat format that can be used for further studies. Some studies have performed transcriptomic and proteomic analysis of *T. pesudonana* under different conditions [53,54,55]. The model can be integrated with omics data (such as transcriptomics or proteomics), which would provide further constraints to the fluxes, thus narrowing the FBA solution space and improving the prediction of the simulated flux distribution. Finally, ^13^C-MFA can be performed in different conditions to validate the predicted FBA solutions.

## 5. Conclusions

This report presents the first publicly available, comprehensive, and manually curated genome scale metabolic model of *Thalassiosira pseudonana*. Several results obtained by the model were found to be in agreement with previous reports. The model was used to probe the metabolic engineering needed to produce various native and heterologous industrially relevant compounds that can be produced by the organism. The model will be a useful tool for systems-level investigation of *Thalassiosira pseudonana*.

## Figures and Tables

**Figure 1 microorganisms-08-01396-f001:**
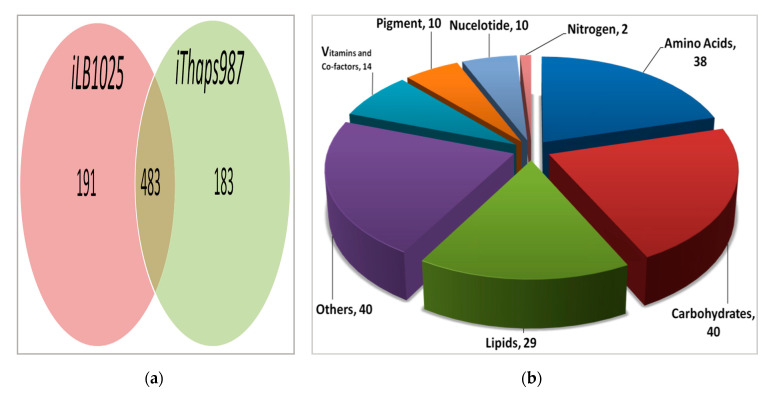
Comparison of two diatom GEMs. (**a**) A Venn diagram showing common and unique enzymes in *i*Thaps987 and *i*LB1025 models. (**b**) Distribution of unique enzymes *i*Thaps987 in various main pathway categories. (**c**) Distribution of unique enzymes in sub-pathways of main pathway categorie*s.* The color coding follows the colors used for the corresponding pathway in (**b**).

**Figure 2 microorganisms-08-01396-f002:**
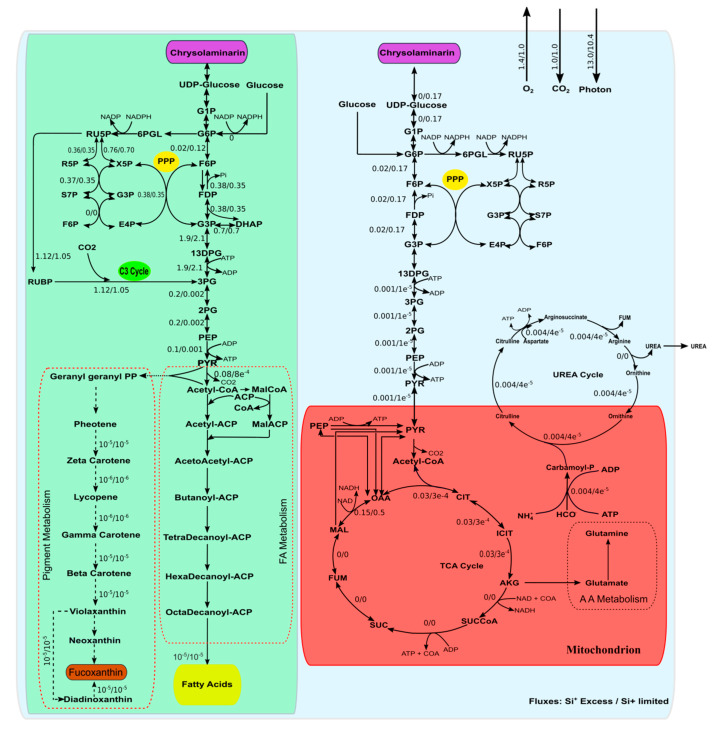
Flux-map showing flux distributions under Si-rich/Si-limited condition. Chloroplast are shown in green color while mitochondria are shown in red. Photosynthesis, lipid, and fucoxanthin synthesis occurs in chloroplast while the TCA cycle and part of the urea cycle occurs in mitochondria. Other major pathways are located in cytosol. The flux through pathways was simulated by fixing the measured growth rate and using the minimization of total flux as the objective function.

**Figure 3 microorganisms-08-01396-f003:**
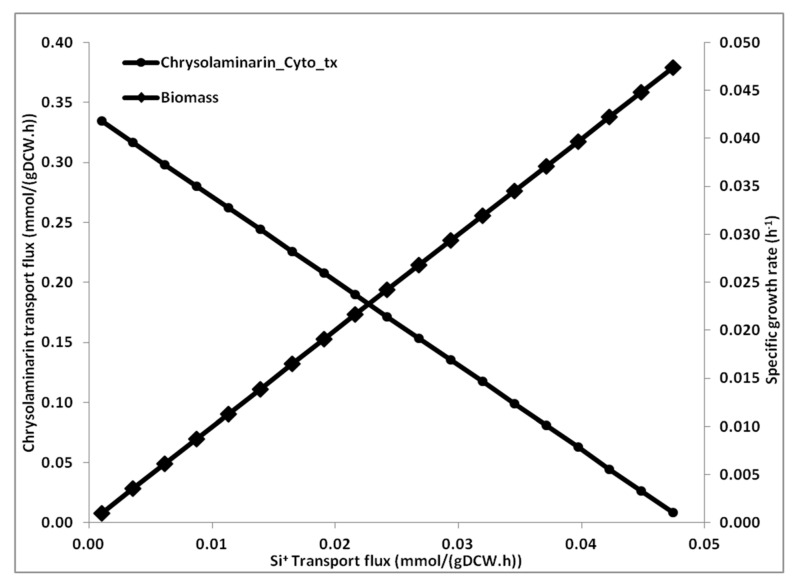
Effect Si^+^ transport variation on biomass and chrysolaminarin production. For these simulations, the Si^+^-transport was varied from zero to the value obtained under Si^+^ replete condition (0.05 mmol/(gDCW.h) and the fluxes through biomass and chrysolaminarin biosynthesis reactions are presented.

**Figure 4 microorganisms-08-01396-f004:**
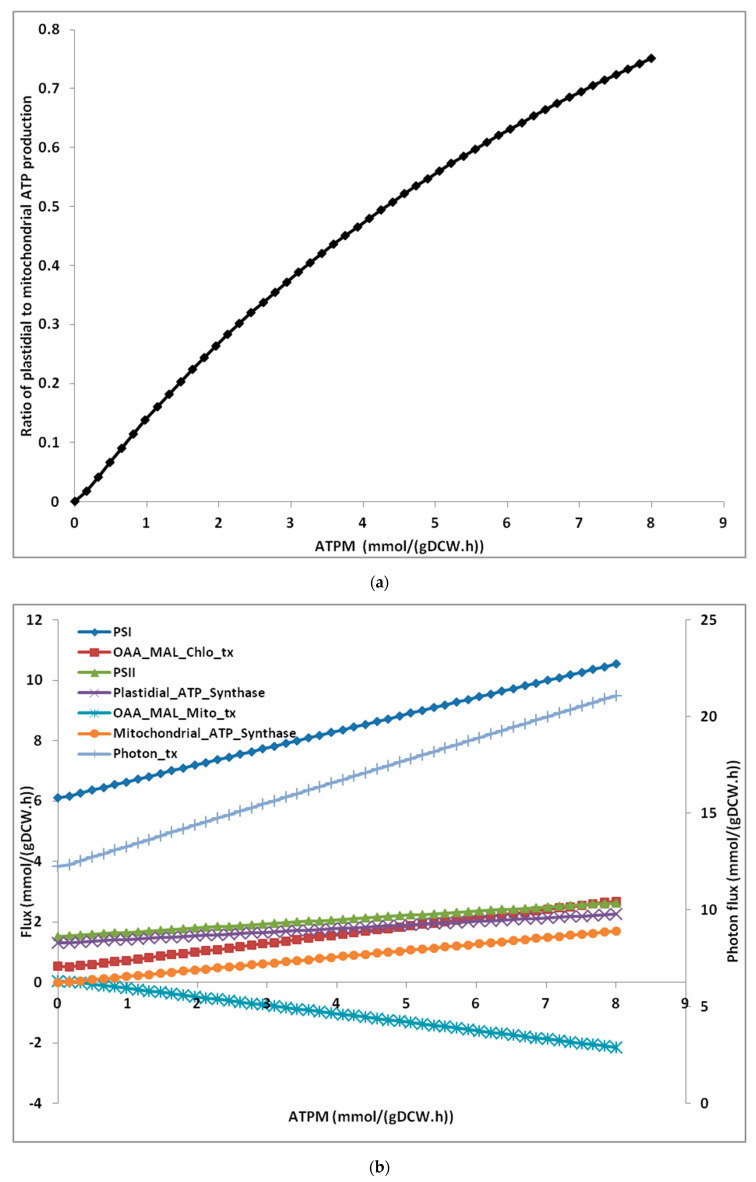
Simulated effect of variation of maintenance ATP (ATPM) values on metabolism. (**a**) Effect of ATPM values on the ratio of mitochondrial and plastidial ATP synthesis. The flux through ATPM reaction was varied and the effect on plastidial and mitochondrial ATP synthesis reactions was simulated. (**b**) Effect of ATPM values on flux through different reactions of the model, *i*Thaps987. PSI: Photosystem-I, PSII: photosystem-II, Photon_tx: Photon transport, Plastidial_ATP_Synthase: Plastdial ATP synthesis reaction, Mitochondrial_ATP_Synthase: Mitochondrial ATP synthesis reaction, OAA_MAL_Chlo_tx: Malate–oxaloacetate shuttle_chloroplast_transport, OAA_MAL_Mito_tx: Malate–oxaloacetate shuttle_ mitochondria_transport. (**c**): The figure shows the fluxes under low (1.5 mmol/(gDW.h)) and high (7.5 mmol/(gDW.h)) ATP maintenance values. The values written in green were obtained from simulations performed under normal wild-type conditions while the values written in red were obtained from simulations performed by knocking out the transporter for mitochondrial malate–oxaloacetate shuttle. LEF: Linear Electron Flow, CEF: Cyclic Electron Flow, PSI: Photo-system I, PSII: Photo-system II, PQ: Plastoquinone, PC: Plastocyanin, Fdxn: Ferredoxin.

**Figure 5 microorganisms-08-01396-f005:**
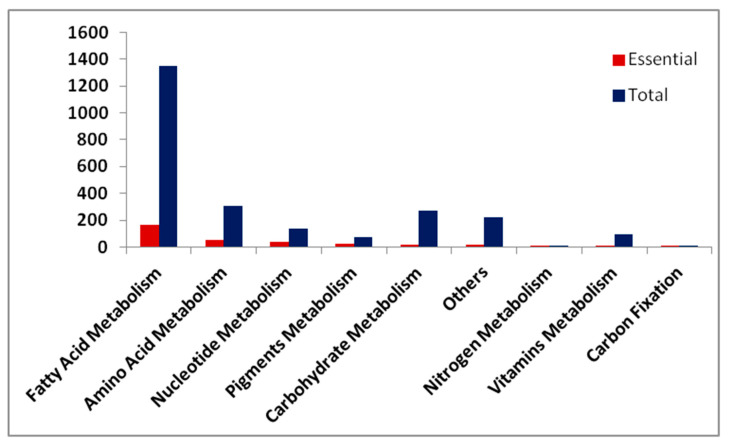
Distribution of total and essential reactions of the model across various subsystems.

**Table 1 microorganisms-08-01396-t001:** Distribution of reactions and metabolites in different compartments of the model *i*Thaps987.

	Cytosol	Chloroplast	Mitochondrion	Transport
Reactions	1125	887	377	88
Metabolites	1024	984	467	-

**Table 2 microorganisms-08-01396-t002:** Comparison of the *i*Thaps987 model with other published models of diatoms.

Microorganism	*P. tricornutum*	*P. tricornutum*	*T. pseudonana*
Genome size (Mbp)	27.45	27.45	32.61
Genes in the model	1027 (10,392)	-	987
Number of reactions/metabolites	2156 */2172	849/587	2477/2456
Compartments	6	4	3
Reference	[28]	[29]	This study

* Levering et al. had given two models in the same publication; one is normal model with 2156 reactions and the other model is lipid metabolism specific model having 4456 reactions.

**Table 3 microorganisms-08-01396-t003:** Simulated maximum theoretical yields of various native and non-native products under photoautotrophic conditions.

S.N	Product	Yield Photoautotrophic (mol/mol carbon)	No. of Added Reactions	Added Genes
	**Native**	*i*Thaps987		
1	Malate	0.22	0	−
2	Succinate	0.22	0	−
3	Citrate	0.14	0	−
	**Non-Native**			
4	2-MethylSuccinate	0.17	4	*cimA*, *leuC*, *leuD*, *ER*
5	Styrene	0.11	2	(*PAL*, *PAL1*, *encP*), *FDC1*
6	iso-butanol	0.22	1	*kivD*
7	Valencene	0.06	2	*Valcs*, *Vals*, *Tps1*
8	Farnesyl-PP	0.06		*afs*
9	Naringenin	0.06	3	*PAL*, (*CHI1* and *CHI2*),*CHS-A*
10	5-Amino-Levulinate	0.17	1	*hemA* or *HemT*
11	Isoprene	0.17	4	*Isps*, *IspG*, *IspH* * (* catalyses 2 reactions)
12	PHB	0.22	2	*phbB*, *phbC*
14	Citramalate	0.17	1	*cimA*

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
