# Peer review of "A Genome-Scale Metabolic Model of *Thalassiosira pseudonana* CCMP 1335 for a Systems-Level Understanding of Its Metabolism and Biotechnological Potential"

_microorganisms, 2020, doi:10.3390/microorganisms8091396_

Round 1

Reviewer 1 Report

This is a very interesting paper, in which a genome-scale model of diatom T. pseudonana was constructed and reported the characteristics.

Comment 1: The supplemental file S02_iThaps987.xls does not show the functional classification of each reaction. It is difficult to understand what reaction the means anything to the reader. When you us to display the functional classification will help the reader to understand.

Comment 2: Is only the MAL-OAA shuttle described in the GMM for the exchange of substances between mitochondria and chloroplasts? Any other shuttle (eg malate-aspartate shuttle)? If not mentioned, is the MAL-OAA shuttle as important as it is?

Comment 3: I couldn't find how the reaction of PSI was described. Is photons up-take ratio between PSII and PSI always set to 1:1? Or is it free? If it can be taken freely, it seems that the photon uptake ratio of PSI/PSII is set so as not to drive the path for flowing excess electrons such as CEF in FBA simulation. Is there data such as PSI/PSII ratio of T. pseudonana?

Comment 4: The metabolism of T. pseudonana is C3 plant-like and the function of the MAL-OAA shuttle is described. Is it possible to experimentally prove the metabolic flux distribution prediction by these FBA? I think that verification by 13C-MFA is necessary to prove the accuracy of simulation prediction.

Reviewer 2 Report

The article entitled “A genome-scale metabolic model of Thalassiosira pseudonana CCMP 1335 for a systems-level understanding of its metabolism and biotechnological potential” by Ahmad et al. presents a genome-scale metabolic model of T. pseudonana. The authors analyzed variations in the metabolism of T. pseudonana under many in silico conditions. The choice of analyses is appropriate to describe the metabolism of a microorganism. However, this study contains several drawbacks that should be addressed before publication. Please see below some comments, which may help to improve the manuscript:

Major comments:

The main shortcoming of the study is that there are no data provided, which can prove that the model has been validated using experimental data (growth and other physiological data). This should be a minimal requirement for any modeling work – otherwise the accuracy of the model and its use are difficult to estimate. Experimental measurements can be employed from previous studies.

Line 90 – The authors mentioned that the biomass composition was measured to construct the biomass reaction. Why was this data not provided in the manuscript or in the supplementary files? Detailed biomass measurements are hard to come by and this would be a great resource to have. Also, the details of how the biomass reaction was reconstructed, is missing in the article.

Model and supplementary files should be provided for review. Without these materials, the quality and significance of predictions cannot be checked!

Some figures need major improvement. The authors need to provide a caption explaining the figure and its labels. Some figure panels are shown as separate figures, please merge them into one figure with a unique title, and legend.

Figure 1: The abbreviations in the labels are not explained anywhere. The colors need to be lighter to make the labels easy to read. Moreover, the three-dimensional display is unnecessary and the overall figure looks too big for the amount of information that is conveyed. Please choose a simpler and more compact form of representation of model properties.

Figure 2a: Similar to Figure 1, the use of space is not efficient. The pie chart is too big for the amount of information that is shown. This panel can be made smaller or complemented with a comparison of unique enzymes in iThaps987 and iLB1025 in, for example, a Venn diagram.

  1. How was the analysis shown in Figure 4 performed? Were biomass production and chrysolaminaran production maximized? Please elaborate in the methods section.

Minor comments:

Line 42 – The symbol should be corrected.

Lines 71-80 – These three small paragraphs can be merged into one.

Lines 305-310 – Please cite some studies to support this explanation.

Figure 3 – Choose lighter colors. Especially, please replace the red color by any light color. A part of the network is not visible.

Figure 4 – These predictions should be supported by previous related studies. 

Lines 440-442 – reference is needed.

In line 51, replace "complete" with "compete". Moreover, it is not clear what this refers to. In what settings will it not compete? Are you referring to industrial cultures for lipid/biofuel production?

Lines 63 to 65, the idea that they belong to two different groups is repeated.

Line 62 to 80, why is the arrangement of fustrules in the cell wall relevant to discard P. tricornutum as a model for the metabolism T. pseudonana? Use a diatom study to provide some examples on key differences between both to better support this claim.

Line 89, replace "centre" with "Center"

Line 92 to 93, this idea is awkwardly worded and confusing, please rephrase.

Line 98, define PGDB in the first use.

Line 106, remove or rephrase "high quality model is obtained"

Line 138, provide some examples of assumptions in the cited studies that are relevant to support your assumption.

Line 155, explain here why was 0.024 1/h the chosen growth rate.

Line 320, a previous study (Smith et al., Nat Commun. 2019;10.) suggests that the urea cycle is split into an anabolic and a catabolic segment in P. tricornutum. The former is used for arginine synthesis, while the latter is used for arginine degradation under nitrogen starvation. Is this happening in T. pseudonana as well?

Line 437 to 439, this paragraph should be complemented. What other similarities or differences were found? 

Round 2

Reviewer 2 Report

I reviewed a previous version of this manuscript and asked a) the model to be provided and b) several corrections to improve readability of this work.

The figure legends are still too brief and cryptic; they do not provide enough information about each panel. It is recommended that the authors should check the maximum word limit for figure legends and use this space properly to explain each figure.

The model is of extremely poor quality. Based on that quality, the present version of the will not benefit any future users.  

The quality of the model should be checked based on the previously published protocol (https://www.nature.com/articles/nprot.2009.203). After improving all the quality parameters, the author should re-run the simulations to check if the model predictions are consistent or more manual curation steps are required.

For example, (i) the model contains many duplicate reactions, mostly one reaction is reversible, and the duplicate one is irreversible. This is a very basic and critical check for the accuracy of the model.

Reaction set 1: (see the duplicate set below)

'G3PDH_Chlo'  

'PYRNUTRANSHYDROGEN-RXN_Cyto'  

'PHOSPHOGLUCMUT-RXN_Cyto'  

'ACETYL-COA-CARBOXYLTRANSFER-RXN_Cyto'  

'SUCCINATE-DEHYDROGENASE-UBIQUINONE-RXN_Mito'  

'GLYC3PDEHYDROGBIOSYN-RXN_Cyto'  

'GLYK_Cyto'  

'RXN-11662_Mito'  

'MALATE-DEH-RXN_Mito'  

'MGDGS_HDE_HDE_Chlo'  

'PAPA_PALM_PALM_Chlo'  

'PAPA_HDE_HDE_Chlo'  

'RXN-6182_Cyto'  

'RXN-7978_Chlo'  

'RXN-7979_Chlo'  

'RXN0-6377_Cyto'

Reaction set 2:

'G3PDH_Chlo'  

'PYRNUTRANSHYDROGEN-RXN_Cyto'  

'PHOSPHOGLUCMUT-RXN_Cyto'  

'ACETYL-COA-CARBOXYLTRANSFER-RXN_Cyto'  

'SUCCINATE-DEHYDROGENASE-UBIQUINONE-RXN_Mito'  

'GLYC3PDEHYDROGBIOSYN-RXN_Cyto'  

'GLYK_Cyto'  

'RXN-11662_Mito'  

'MALATE-DEH-RXN_Mito'  

'MGDGS_HDE_HDE_Chlo'  

'PAPA_PALM_PALM_Chlo'

'PAPA_HDE_HDE_Chlo'  

'RXN-6182_Cyto'  

'RXN-7978_Chlo'  

'RXN-7979_Chlo'  

'RXN0-6377_Cyto'

(ii) Reactions are neither mass balanced nor charge balanced. This is a absolute must to assure model accurary!

(iii) Metabolite charge information (model.metCharge) is incorrect.

(iv) The model does not contain the gene list and gene-reaction associations (grRules). This means the model cannot be used for any gene knockout investigations! The model cannot even relate to the genome sequence.

(v) There are other empty fields in the model: model.c, model.metChEBIID, model.metCompartment, model.rxnGeneMat etc..

3. The author must complete all the fields in the model and improve the quality, as suggested in the above comment. After the model is massively improved, check model quality using MEMOTE (https://memote.io/; https://www.nature.com/articles/s41587-020-0446-y) and incorporate one more figure to the main article showing different quality scores of the model.
